# Influence of Milling Time on Phase Composition and Product Structure of Mg-Zn-Ca-Ag Alloys Obtained by Mechanical Synthesis

**DOI:** 10.3390/ma15207333

**Published:** 2022-10-20

**Authors:** Sabina Lesz, Małgorzata Karolus, Adrian Gabryś, Marek Kremzer

**Affiliations:** 1Department of Engineering Materials and Biomaterials, Silesian University of Technology, 18a Konarskiego St., 44-100 Gliwice, Poland; 2Institute of Materials Engineering, University of Silesia, 1a 75 Pułku Piechoty St., 41-500 Chorzów, Poland; 3Nanotechnology and Materials Technology Scientific and Didactic Laboratory, Silesian University of Technology, 7a Towarowa St., 44-100 Gliwice, Poland

**Keywords:** metallic alloys manufacturing, Mg-based alloy, mechanical alloying, SEM, XRD

## Abstract

Magnesium-based alloys are widely used in the construction of automotive, aviation, and medical applications. The solutions presently used for the production of biodegradable materials are characterized by considerable energy consumption and limitations resulting from the use of different devices and technologies. The proposed material is easier to manufacture due to mechanical alloying (MA). Thanks to the MA process, it is possible to carefully tailor the desired chemical composition and microstructure. There are many parameters that can be modified during synthesis in order to obtain an alloy with the desired microstructure and specific expected alloy properties. The duration of grinding of the alloy, the size and number of balls, and the protective atmosphere have a great influence on the process of mechanical alloying and the properties of the obtained products. So, the aim of this work was to determine the influence of milling time on the phase composition and structure of Mg-based alloy synthesis products. The tested samples were milled for 5, 8, 13, 20, 30, 50, and 70 h. X-ray diffraction analysis (XRD) and scanning electron microscopy studies (SEM) with energy-dispersive spectroscopy (EDS) were performed to obtain the powder morphology and chemical composition of Mg_66−x_Zn_30_Ca_4_Ag_x_ (where x = 1, 2) powders. Structure characterization based on the Rietveld refinement and crystallite size determination based on the Williamson–Hall theory of milling products were also carried out.

## 1. Introduction

Magnesium alloys have been used for many years, mainly in the automotive industry, due to their low density and favorable properties. Currently, they are of interest to many research centers, due to their potential applications in the field of biomedical engineering. The biomaterials used so far for metal implants are austenitic steels, cobalt- and titanium-based alloys, and precious metals. They are characterized by a favorable set of mechanical properties and resistance to corrosion. However, after a certain time, the implant becomes toxic due to the reactions of the elements present in it with body fluids and tissue. Therefore, it becomes necessary to re-operate to remove the implant. Therefore, magnesium-based alloys, which are biodegradable and bring many advantages over traditional permanent implants, find potential for medical applications. So far, technologies for the production of biodegradable metal materials have been used, including casting processes and processing by solid-state shaping by means of metalworking and machining. These technologies are well known but require a complicated introduction of a protective atmosphere during the melting process. It is of paramount importance to find new manufacturing techniques that will produce the unique microstructure of magnesium-based alloys with controlled corrosion characteristics. The idea is to produce magnesium alloys by mechanical alloying (MA—mechanical alloying) and then spark plasma sintering of powders by the SPS method (SPS—spark plasma sintering).

One of the basic groups of the new generation of materials in the modern world is functional materials [1,2,3,4,5,6,7,8,9,10,11,12,13]. Among them, great progress is being made in the field of materials with special physicochemical properties, especially biomaterials. Biomaterials are a specific group of materials with different compositions, structures, and properties, distinguished by a specific feature, namely that they are accepted by the human body—its regeneration [1]. Based on the definition of the European Society for Biomaterials [2], a biomaterial is any substance that is not a drug or a combination of substances (natural or synthetic) that is used to heal or replace tissues of an organ or function. Attempts to implant foreign materials into human tissue practically date back to the beginnings of medicine. Various materials have been used, including wood, animal bones, and precious metals (gold, silver) [1,2,3,4,5,6,7,8,9,10,11].

Biomaterials are made of metals, ceramics, plastics, glass, living cells, and tissues. These materials can be made into molded or machined parts, coatings, fibers, films, foams, and fabrics for use in biomedical products and devices. These can include heart valves, hip replacements, dental implants, or contact lenses. They can also be biodegradable, and some are bioresorbable, which means that they decompose in a controlled manner in the body into inert, non-toxic molecules and are absorbed after fulfilling their function [1,5,14].

Mg is a silvery-white metal with a density of 1.738 g/cm^3^ (at room temperature), which, for liquid magnesium (melting point of 650 °C), is 1.585 g/cm^3^. It oxidizes very easily in contact with air, creating a protective layer of oxides. The flash point is below 500 °C, the boiling point is 1090 °C, and the flame is approximately 2200 °C. It is flammable in fragmentation (foil, powder). While burning, it creates a bright white light. It reacts with water at room temperature, releasing hydrogen [15,16].

Pure Mg has low mechanical properties and high chemical activity, therefore it is not used as a construction material. These properties can be improved by creating magnesium alloys [17,18,19].

Magnesium alloys are, along with stainless steel, titanium, cobalt, and chromium alloys, one of the most popular metallic materials used in biomedical applications. Magnesium-based alloys are light, with a density similar to cortical bone, and much lighter than stainless steel, titanium alloys, or chrome-cobalt alloys. Magnesium’s modulus of elasticity is similar to that of natural bone, which reduces the risk of stresses arising with other parts of the body. Magnesium alloys are highly biocompatible and non-toxic with the appropriate alloy additions [20].

As alloying additions to magnesium, the following can be used [21]:Zinc (Zn), which is biocompatible, when added to magnesium, creates a dispersive phase, enabling precipitation hardening.Calcium (Ca), which is biocompatible, improves fluidity, and refines the grain, and when added to the Mg-Zn system, increases hardness and creep resistance. The addition of Ca increases the hardness of alloys, which is related to precipitation hardening.Precious metals, for example, silver, cause the formation of a passive layer, improving corrosion resistance, or gold, which improves corrosion resistance and increases strength and hardness [20,21,22,23,24,25,26,27].

Magnesium-based biodegradable implants can dissolve, be absorbed, or be excreted by the human body after the tissue heals. This shortens patients’ convalescence and reduces the cost of treatment. The composition of the magnesium alloy must be selected so as not to introduce toxic substances (elements) into the body. The low corrosion resistance of magnesium and calcium, especially in an electrolytic aqueous environment, is a major disadvantage in engineering applications; however, it becomes useful for biomaterials used for implants [28].

The Mg-Zn-Ca alloy can be used as a medical implant; however, it is necessary to improve its mechanical properties, especially plastic properties. The value of the longitudinal elasticity modulus of this alloy (approximately 30–35 GPa) is comparable with the longitudinal elasticity modulus of the cortical tissue (approximately 2–20 GPa) [29].

Among the Mg-Zn-Ca alloys tested for corrosion resistance, the highest cumulative volume of released hydrogen 1.93 mL/cm^2^ (after two weeks of immersion in Ringer’s solution) was shown by the Mg_68_Zn_28_Ca_4_ alloy, the lowest by 1.21 mL/cm^2^ by the Mg_64_Zn_32_Ca_4_ alloy, and Mg_66_Zn_30_Ca_4_ showed 1.52 mL/cm^2^. During the research, it was observed that during the first 3–4 h of immersion in Ringer’s solution, the most hydrogen is released from Mg alloys [29].

It was confirmed by Zberg et al. [30] that when the Zn content in the alloy is higher than 28% (at.), its corrosion resistance is significantly improved. This is due to the formation of a passive oxide layer, the presence of which has been confirmed by the results of X-ray dispersion. Zinc is a particularly important element limiting the evolution of hydrogen. An alternative method to the classically used metallurgy for the fabrication of implants is powder engineering (PE). One of the methods of powder manufacturing is mechanical synthesis, also known as mechanical alloying (MA).

The technology for the production of powder materials used so far is characterized by high energy consumption and is limited in the selection of substrates (e.g., only highly soluble salts). A method that eliminates these limitations is mechanochemical treatment, based on, e.g., the use of frictional forces to generate physicochemical changes in materials. During the grinding process, very large amounts of mechanical energy are supplied to the material, which is accumulated in the form of lattice stresses and/or converted into chemical energy. As a result, structural changes in the material appear (defects, dislocations, amorphous phases), and/or chemical reactions take place. The effect inherent in the mechanical processing of the material is its fragmentation and increased physical and chemical activity.

Mechanical alloying (MA) is the process of grinding a substance in a solid state to obtain the alloy in powder form. The feedstock is the pure element in appropriately selected proportions or alloys that are subjected to the grinding process in high-energy mills. The material is placed in steel or ceramic tanks filled with a grinder made of appropriate materials, such as tungsten carbides, hardened steel, ceramics, etc. The process produces particles of a similar size. This method is used for the amorphization of alloys, but also for grinding the base material and changing the microstructure, as well as creating intermetallic phases. As a result of the process, the input material is crushed and agglomerated. Due to cyclic deformation, i.e., melting, crushing, and remelting, the grain size is reduced, and new grain boundaries are created. The structure of the material is not stable, and the alloy may exist as a solid solution, an intermetallic phase, a mixture of components, or an amorphous material [31,32,33,34].

Through mechanical synthesis, it is possible to obtain materials that cannot be produced by classical methods, such as casting, as well as materials with a strictly defined chemical composition and high purity. The process of mechanical synthesis with the above-mentioned features enables the production of components, including biomedical implants with complex shapes and high-entropy alloys [34,35,36].

The benefits of powder metallurgy (PM) over conventional casting methods are related to the control of porosity, pore size, and pore distribution [37,38].

## 2. Materials and Methods

Based on the literature review, it was found that magnesium-based alloys enriched with noble metals (Noble Metal—NM) are a promising material for biomedical applications. For the purposes of this study, the addition of Ag not exceeding 2 at.% was selected. Finally, among the alloys from the Mg_66−x_Zn_30_Ca_4_NM_x_ system (where NM—noble metal: Ag; x = 1, 2), alloys and sinters with the atomic composition of Mg_65_Zn_30_Ca_4_Ag_1_ and Mg_64_Zn_30_Ca_4_Ag_2_ were produced and tested.

The following Alfa Aesar powders were used in the process of mechanical synthesis:Zinc (Zinc powder)—100 mesh, 99.9% (metal basis).Magnesium (Magnesium powder)—20 + 100 mesh, 99.8% (metal basis).Calcium (Calcium shot), redistilled, 1 cm & down, 99.5% (metal basis).Silver (Silver powder)—100 mesh, 99.95% (metal basis).

For the mechanical synthesis, the elements of the alloys, i.e., Mg, Zn, Ag, and Ca, were weighed as described in Table 1. They were placed in a steel crucible (Figure 1) containing 23 steel grinding balls. Based on the test results [29,38,39,40,41], the ball-to-powder ratio (bpr) was established at 10:1. The 10 mm balls were made of 316 L stainless steel. In order to protect the powders from oxidation, the mill charge was prepared under a protective gas—argon. The mechanical synthesis process was performed using an 8000D Mixer/Mill-type high-energy ball mill (SPEX SamplePrep, Metuchen, NJ, USA—Figure 2). The duration of grinding was 5 to 70 h. Each grinding process consisted of 1 h of grinding and a 0.5 h break.

This paper presents the results of the structural analysis of the obtained Mg_66−x_Zn_30_Ca_4_Ag_x_ alloys (where x = 1, 2) (X-ray diffraction, Rietveld analysis), microstructure analysis (scanning microscopy SEM with EDS, transmission electron microscopy TEM), and particle size [42]. X-ray diffraction measurements of the obtained alloys were made with an Empyrean diffractometer (PANalytical, Almelo, The Netherlands) using a lamp with a copper anode (Cu Kα = 1.5417 Å) and a PIXCell counter, using the step-scanning method in the range of angles from 10 to 100° 2θ. The phase analysis of the alloys was performed using the High Score Plus PANalytical software (version 4.0, PANalytical, Almelo, Netherlands) and the ICDD PDF4 + 2016 database (International Center for Diffraction Data, Newtown Square, PA, USA). The structural characteristics of the alloys and crystallite sizes were carried out using the Rietveld method [43,44,45,46] implemented in the High Score Plus PANalytical software. Calculations of the parameters of unit cells of the identified phases and the determination of their crystallite size and lattice strains were performed using the Rietveld analysis, which is based on modifying the assumed crystallographic structure and analyzing the profile of the registered diffraction lines [43,44]. The quality of the match is determined by the so-called R-parameters, which, in relation to the presented results, reached acceptable values below 10%. According to the literature, the generally accepted accuracy of such determined lattice parameters is at the level of 10^−5^ Å. The accuracy of crystallite size estimation, according to the Williamson–Hall theory, included in the Rietveld procedure, is 10–15%.

Microstructure analysis was carried out using a JEOL JEM-3010 high-resolution transmission electron microscope (TEM, JEOL Ltd., Tokyo, Japan) with 300 kV acceleration voltage, equipped with a Gatan 2k × 2k Orius™ 833 SC200D CCD camera (Gatan Inc., Pleasanton, CA, USA). The powder sample was suspended in isopropanol, and the resulting material, after dispersion in an ultrasonic bath, was deposited on a Cu grid with an amorphous carbon film standardized for TEM observations.

The particle size distribution of the alloys from the Mg_66−x_Zn_30_Ca_4_NM_x_ system (where x = 1, 2) was determined using the Analyssette 22 MicroTec + apparatus (Fritsch, Weimar, Germany) by performing measurements in ethyl alcohol.

## 3. Results and Discussion

### 3.1. XRD Analysis

The Mg_66−x_Zn_30_Ca_4_Ag_x_ alloys (where x = 1, 2 at.%) were obtained using the mechanical synthesis (MA) process in a ball mill. The studies analyzed alloys with silver content in the amount of 1 and 2 at.%. The samples were milled for 5, 8, 13, 20, 30, 50, and 70 h.

The analysis of the diffraction patterns of alloys with silver addition (Figure 1 for Ag1 = 1 at.% and Figure 2 for Ag2 = 2 at.%) showed that the alloy structure contains the following phases: MgZn_2_ and Zn and Mg traces. The amorphization process is slow and was observed only after 50 h of milling for the Ag1 alloy and 70 h of milling for the Ag2 alloy. The diffraction patterns show a gradual disappearance of the diffraction lines of zinc, which dissolves in magnesium to form a hexagonal solid solution of Mg (X), where X represents the alloying additions, i.e., Zn, Ca, and Ag. According to the presented results of the X-ray analysis and the analyses of the chemical composition of the ground powders, the presence of impurities resulting from the abrasion of the material of the spheres and vessel walls was not observed.

In the case of the Ag1 alloy, a gradual process of amorphous material can be observed. The first step in the formation of an amorphous phase after 20 h of grinding leads to intermediate crystalline phases after 30 h to re-amorphization after 50 h of grinding. Similar milling processes are observed in the literature. Exemplary detailed considerations on transitional amorphous forms were presented in research on the mechanical synthesis of Ni-Ti alloys [47].

Amorphization is a progressing process. It is induced both thermally and mechanically. It is already observed in the initial and intermediate stages of grinding. However, after 50 and 70 h, the amorphous phase is the dominant phase in the Ag1 alloy. It was found on the basis of the XRD results, where the disappearance of the reflections originating from the crystalline phases was observed (a drastic decrease in the intensity of the reflections). In the Ag2 alloy, the process is similar, but the disappearance of reflections takes place after 70 h of grinding.

The results of structural parameters, crystallite sizes, and the lattice strain for the main phases obtained for the Ag1 alloy after 5 h MA are presented in Table 2, and the results obtained for the Ag2 alloy after 5 and 30 h of milling are presented in Table 3.

The crystallites size of the main phases observed in the material, i.e., MgZn_2_, and the magnesium-based solid solution Mg(X) (where X = Zn, Ca, Ag), are at the level of 500–550Å, and the values of the unit cell parameters slightly change (Table 2 and Table 3). The amorphization process is observed only in the case of the Mg_65_Zn_30_Ca_4_Ag_1_ alloy milled for 50 and 70 h. Despite the fact that there are no clear changes in the crystallite size and the unit cell parameters of the main phases, it might indicate a stable structure of the obtained material.

### 3.2. SEM Analysis

Using a scanning electron microscope by Zeiss Supra 35, equipped with the EDS—Trident XM4 chemical composition analysis system by Edax, the microstructure of the analyzed alloys was examined, and their chemical composition was determined.

The comparative analysis of SEM images performed for all of the alloys tested showed high similarities in the morphology of the samples. The SEM images show grains of varying degrees of fragmentation depending on the grinding time and areas characterized by the presence of agglomerates.

Figure 3 and Figure 4 show an example of the microstructure (SEM) and the results of the chemical analysis (EDS) of the Mg_65_Zn_30_Ca_4_Ag_1_ alloys after 70 h of milling during the mechanical alloying. The analysis of the microstructure of the Mg_65_Zn_30_Ca_4_Ag_1_ and Mg_64_Zn_30_Ca_4_Ag_2_ alloys obtained in the process of mechanical synthesis and the analysis of their chemical composition using the EDS method show that the material with an initially heterogeneous distribution of elements gradually homogenizes. After 70 h of grinding, the alloys show a homogeneous structure with a grain size of 10–50 µm (individual grains of 100 µm) and an average chemical composition corresponding to the nominal one. Sample results of the microstructure and chemical composition analysis for the Mg_65_Zn_30_Ca_4_Ag_1_ alloy after 70 h of milling are shown in Figure 3 and Figure 4. In the MA process, a new phase with grain boundaries can be formed (area 2 in Figure 4). Chemical analysis shows that the homogeneous grains (1) visible in the images represent the alloy after mechanical synthesis with a chemical composition corresponding to the nominal one, and the grains of the heterogeneous microstructure (2) are richer in zinc with a simultaneous deficiency of Mg and Ca. This may indicate the precipitation of Mg-Zn phases, which is confirmed by X-ray examinations.

As compared to [48], the material presented in Figure 3b resembles a “plate” structure. In the referenced literature, harder particles are processed in the mechanical synthesis, while in the case of this manuscript, silver is much more ductile in comparison. The outcome is different, with a clearer structure as shown in Figure 3. The resulting structure appears due to the processes occurring due to the interaction between softer and harder particles during milling. The harder particles break and decrease in size, and simultaneously, ductile, softer particles are strained and fractured into smaller bits. The structure occurs due to the melting of the softer particles, trapping the harder ones inside and consolidating them. The result is a homogeneous distribution of elements in the particle. Those claims are backed up by the results of the EDS analysis showcased in Figure 4. While Section 1 features a uniform distribution of a smaller particulate after repeated processes, Section 2 of the Figure shows an intermediate process, where the components of the particle are still in a transient phase [35].

The exemplary results of the analysis of the microstructure and chemical composition of the Ag1 alloy after 5 h and Ag2 after 5 h of grinding confirm the occurrence of the transient state, which was revealed in the XRD analysis (Figure 1 and Figure 2, Table 2 and Table 3). The fragmentation of the material indicates the presence of particles with a size ranging from 10–100 µm. Different chemical composition of selected areas (particles) indicates a tendency to agglomeration of alloy components.

The milling processes, as a result of strong plastic deformation, significantly change the morphology of the obtained powders. During mechanical alloying, the same process is repeated many times, i.e., cracking and welding of powder particles. The literature indicates that the milling time may affect the phase composition of the alloys, leading, among others, to the formation of the amorphous phase [49,50] and new crystalline phases. Hence, in addition to the diameter of the spheres and the powder-to-spherical-mass ratio, the grinding time is usually the basic parameter characterizing the course of the mechanical synthesis process.

In the case of the alloys from the Mg_66−x_Zn_30_Ca_4_NM_x_ system synthesized in this work (where: x = 1, 2 at.%), it was found that the short grinding times had essentially no appreciable effect on either the particle size or the phase composition of the alloy.

### 3.3. TEM Analysis

The examples of the TEM image analysis obtained for Mg_65_Zn_30_Ca_4_Ag_1_ alloys milled for 50 h (Figure 5 and Figure 6) and 70 h (Figure 7 and Figure 8) confirm the presence of an amorphous matrix in the tested powders. Figure 5 and Figure 7 present the presence of Debay rings typical of an amorphous phase. Regions with partial alloy crystallization can be observed in Figure 6 and Figure 8, where the electron diffraction images show single reflexes on the Debay rings. The chemical analysis carried out on the tested areas indicates a homogeneous distribution of elements corresponding quantitatively to the assumed composition.

### 3.4. Particle Size

The determined mean particle size of the tested alloys is given in Table 4 and Table 5. In most cases, it can be seen that the statistical number of particles with characteristic diameters (D10, D50, and D90) is comparable for the tested alloys. Deviations from the rule may indicate the formation of larger particles at individual stages of grinding, which are the result of secondary agglomeration and the joining of particles with each other during mechanical synthesis processes (e.g., “Ag2”—after 13, 50, and 70 h).

Figure 9, Figure 10, Figure 11 and Figure 12 show graphs of particle size distributions (histograms) and their cumulative distribution (curve) for selected alloys from the Mg_66−x_Zn_30_Ca_4_NM_x_ system (where NM = Ag and x = 1, 2).

In the graphs showing the particle size distribution for the Mg_64_Zn_30_Ca_4_Ag_2_ alloy after 50 and 70 h of milling (Figure 11 and Figure 12), it can be seen that the material is not homogenous in particle size values. The profiles of the functions describing the histograms are clearly extended and characterized by the presence of two maxima. Undoubtedly, in this case, it is also related to the gradual amorphization of the material and the formation of the nanocrystalline phase.

## 4. Summary

The crystallite size of the main phases observed in the materials, i.e., MgZn_2_ and the magnesium-based solid solution Mg(X) (where X = Zn, Ca, Ag), is at the level of 500–550 Å, and the values of the unit cell parameters slightly changes. The amorphization process is observed only in the case of the Mg_65_Zn_30_Ca_4_Ag_1_ alloy milled for 50 and 70 h. Amorphization is a progressing process. It is induced mainly mechanically. It is already observed in the initial and intermediate stages of grinding. The amorphous phase is the dominant phase in the Ag1 alloy after 50 and 70 h but is dominant in the Ag 2 alloy after 70 h of grinding.

Despite the fact that there are no clear changes in the crystallite size and the unit cell parameters of the main phases, it might indicate a stable structure of the obtained material.

The analysis of the microstructure of the Mg_65_Zn_30_Ca_4_Ag_1_ and Mg_64_Zn_30_Ca_4_Ag_2_ alloys obtained in the process of mechanical synthesis and the analysis of their chemical composition using the EDS method show that the material with an initially heterogeneous distribution of elements gradually homogenizes. After 70 h of grinding, the alloys show a homogeneous structure with a grain size of 10–50 µm (individual grains of 100 µm) and an average chemical composition corresponding to the nominal one.

## Figures and Tables

**Figure 1 materials-15-07333-f001:**
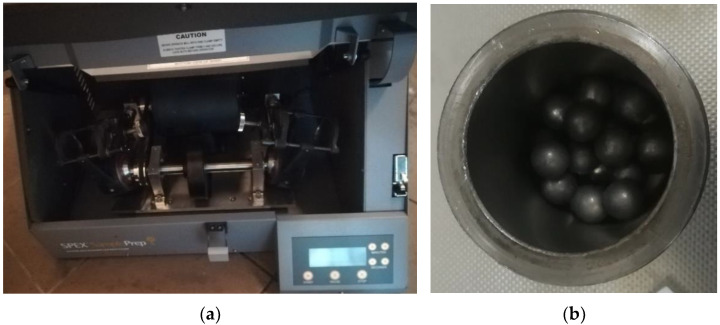
(**a**) High-energy ball mill type 8000D Mixer/Mill; (**b**) steel crucible containing steel balls.

**Figure 2 materials-15-07333-f002:**
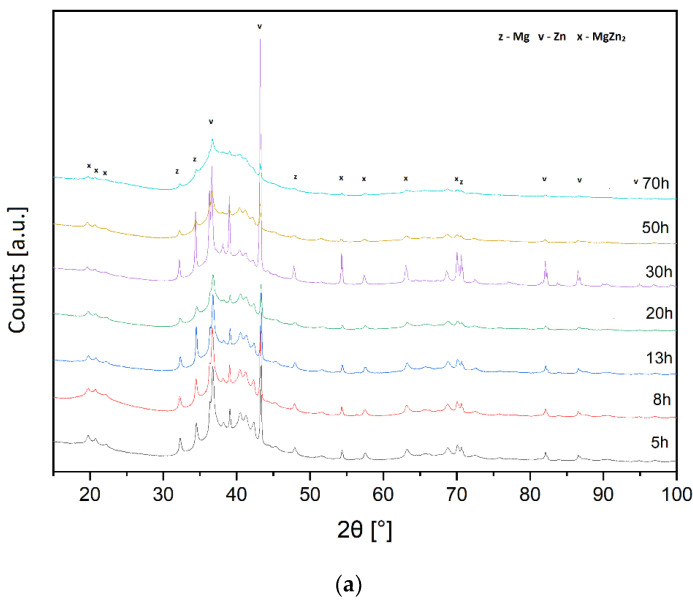
(**a**) The X-ray diffraction patterns of the Ag1 (Ag1 = 1 at.% of Ag) alloy obtained after various milling times (5, 8, 13, 20, 30, 50, 70 h). (**b**) The X-ray diffraction patterns of the Ag2 (Ag2 = 2 at.% of Ag) alloy obtained after various milling times (5, 8, 13, 20, 30, 50, 70 h).

**Figure 3 materials-15-07333-f003:**
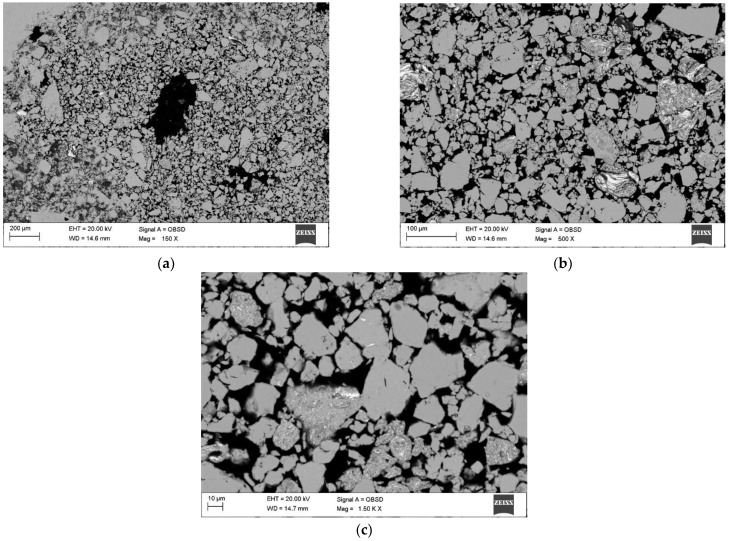
SEM images obtained for Mg_65_Zn_30_Ca_4_Ag_1_ alloy after 70 h of milling, with the following magnification: (**a**) 150×; (**b**) 500×; (**c**) 1500×.

**Figure 4 materials-15-07333-f004:**
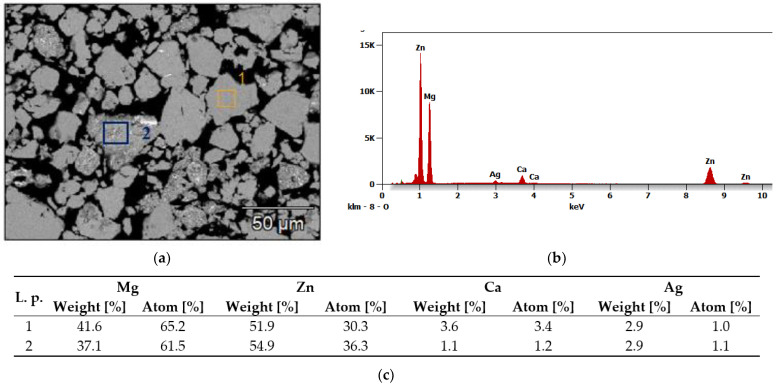
Chemical analysis (EDS) obtained for Mg_65_Zn_30_Ca_4_Ag_1_ alloy after 70 h of milling: (**a**) Analyzed area; (**b**) an example of a chemical composition diagram for area 1; (**c**) chemical composition of selected areas (1 and 2).

**Figure 5 materials-15-07333-f005:**
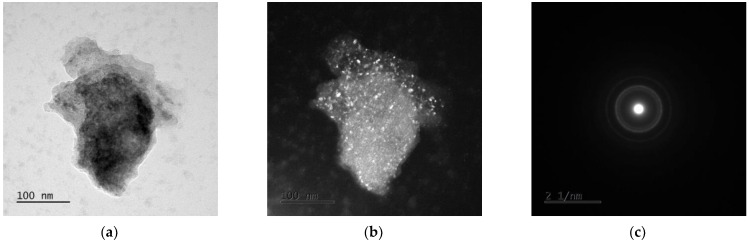
TEM images for Mg_65_Zn_30_Ca_4_Ag_1_ milled for 50 h. (**a**) Bright field image, (**b**) dark field image, (**c**) SAED pattern from tested region.

**Figure 6 materials-15-07333-f006:**
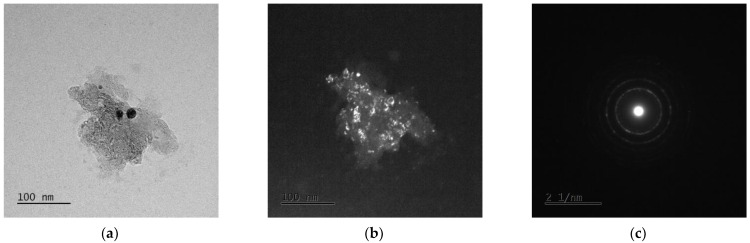
TEM images of Mg_65_Zn_30_Ca_4_Ag_1_ alloy milled for 50 h. (**a**) Bright field image, (**b**) dark field image, (**c**) SAED pattern from tested region.

**Figure 7 materials-15-07333-f007:**
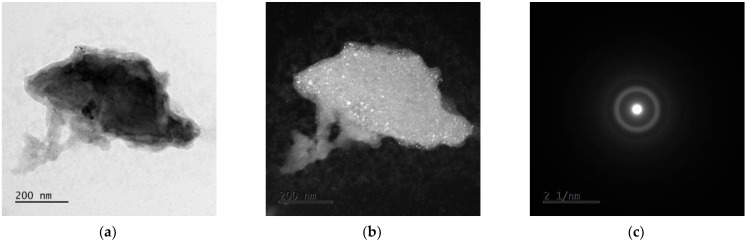
TEM images for Mg_65_Zn_30_Ca_4_Ag_1_ milled for 70 h. (**a**) Bright field image, (**b**) dark field image, (**c**) SAED pattern from tested region.

**Figure 8 materials-15-07333-f008:**
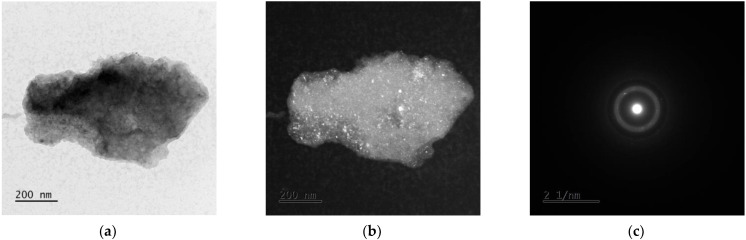
TEM images for Mg_65_Zn_30_Ca_4_Ag_1_ milled for 70 h. (**a**) Bright field image, (**b**) dark field image, (**c**) SAED pattern from tested region.

**Figure 9 materials-15-07333-f009:**
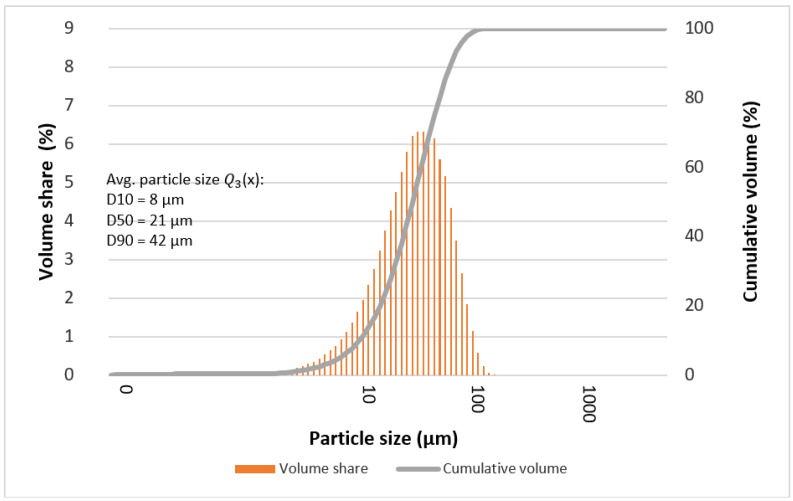
Graphs of particle size distributions (histograms) and their cumulative distribution (curve) for Mg_65_Zn_30_Ca_4_Ag_1_ alloy after 50 milling hours.

**Figure 10 materials-15-07333-f010:**
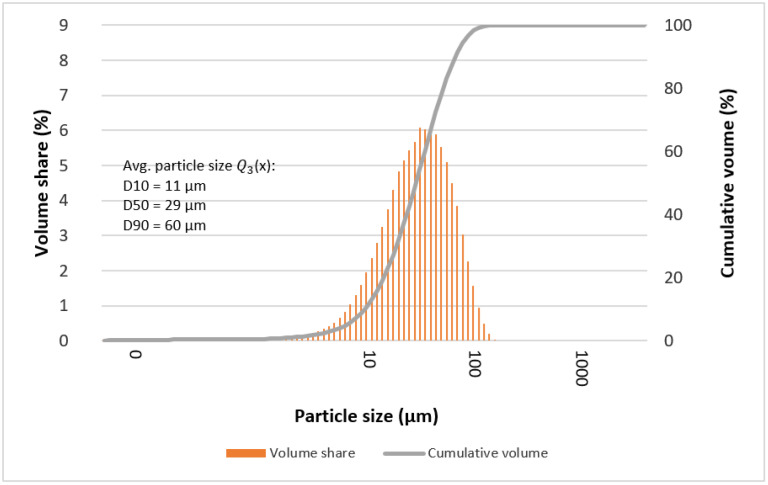
Graphs of particle size distributions (histograms) and their cumulative distribution (curve) for Mg_65_Zn_30_Ca_4_Ag_1_ alloys after 70 milling hours.

**Figure 11 materials-15-07333-f011:**
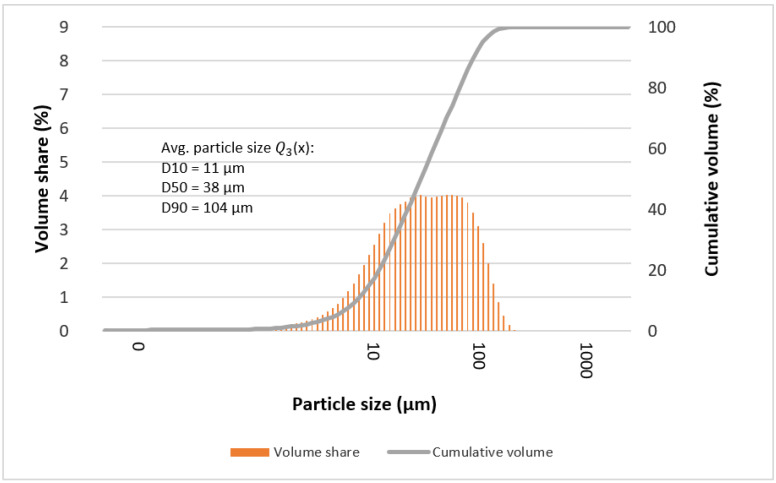
Graphs of particle size distributions (histograms) and their cumulative distribution (curve) for Mg_64_Zn_30_Ca_4_Ag_2_ alloy after 50 milling hours.

**Figure 12 materials-15-07333-f012:**
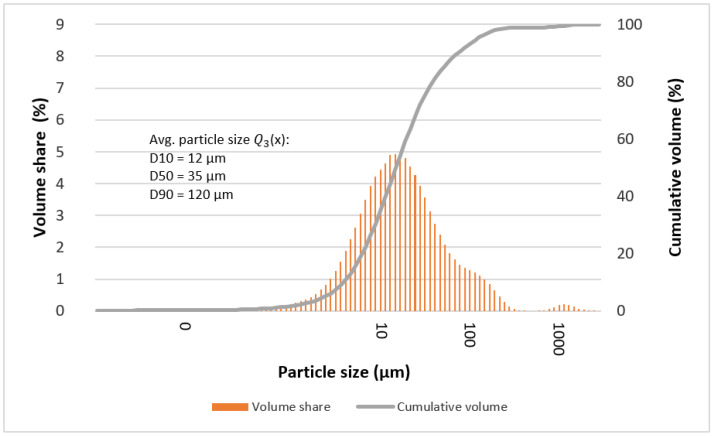
Graphs of particle size distributions (histograms) and their cumulative distribution (curve) for Mg_64_Zn_30_Ca_4_Ag_2_ alloy after 70 milling hours.

**Table 1 materials-15-07333-t001:** Chemical composition (atomic and weight concentration) of alloys (powder weight = 10 g).

Alloy	Mg [g]	Zn [g]	Ca [g]	Ag [g]
Mg_65_Zn_30_Ca_4_Ag_1_	4.147	5.149	0.421	0.283
Mg_64_Zn_30_Ca_4_Ag_2_	3.996	5.038	0.412	0.554

**Table 2 materials-15-07333-t002:** Structural parameters, crystallite sizes, and lattice strain for the main phases present in the Ag1 alloy after 5 h of milling.

Alloy	ICDD PDF4: 00-035-0821 MgZn_2_	Unit Cell Parameters after Rietveld Refinement [Å]	Crystallite Size D [Å]	Lattice Strain η [%]	ICDD PDF4: 04-008-7744 Mg (X) (Where X = Zn, Ca, Ag)	Unit Cell Parameters after Rietveld Refinement [Å]	Crystallite Size D [Å]	Lattice Strain η [%]
**Ag1** **5 h**	a = 3.2050 [Å] c = 5.2150 [Å] Space Group: P63/mmc	a = 3.2040 (5) c = 5.2011 (9)	544	0.47	a = 5.2120 [Å] c = 8.6000 [Å] Space Group: P63/mmc	a = 5.2632 (9) c = 8.7770 (1)	500	0.03

**Table 3 materials-15-07333-t003:** Structural parameters, crystallite sizes, and lattice strain for the main phases present in the Ag2 alloy after 5 and 30 h of milling.

Alloy	ICDD PDF4: 00-035-0821 MgZn_2_	Unit cell Parameters after Rietveld Refinement [Å]	Crystallite Size D [Å]	Lattice Strain η [%]	ICDD PDF4: 04-008-7744 Mg(X) (Where X = Zn, Ca, Ag)	Unit cell Parameters after Rietveld Refinement [Å]	Crystallite Size D [Å]	Lattice Strain η [%]
**Ag2** **5 h**	a = 3.2050 [Å] c = 5.2150 [Å] Space Group: P63/mmc	a = 3.2086 (9) c = 5.2115 (4)	550	0.21	a = 5.2120 [Å] c = 8.6000 [Å] Space Group: P63/mmc	a = 5.2399 (5) c = 9.0966 (4)	500	0.03
**Ag2** **30 h**	a = 3.2050 [Å] c = 5.2150 [Å] Space Group: P63/mmc	a = 3.2021 (1) c = 5.1916 (7)	500	0.82	a = 5.2120 [Å] c = 8.6000 [Å] Space Group: P63/mmc	a = 5.3245 (1) c = 8.7726 (8)	400	0.03

**Table 4 materials-15-07333-t004:** Particle size (µm) for Mg_65_Zn_30_Ca_4_Ag_1_ alloys.

Mg_65_Zn_30_Ca_4_Ag_1_							
Hours	5	8	13	20	30	50	70
D10	14	8	8	8	7	8	11
D50	36	21	27	34	32	21	29
D90	70	41	61	65	60	42	60

**Table 5 materials-15-07333-t005:** Particle size (µm) for Mg_64_Zn_30_Ca_4_Ag_2_ alloys.

Mg_64_Zn_30_Ca_4_Ag_2_							
Hours	5	8	13	20	30	50	70
D10	9	13	12	13	9	11	12
D50	52	37	35	39	28	38	35
D90	76	89	108	91	75	104	120

## Data Availability

Not applicable.

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
