# Peer review of "Influence of Milling Time on Phase Composition and Product Structure of Mg-Zn-Ca-Ag Alloys Obtained by Mechanical Synthesis"

_materials, 2022, doi:10.3390/ma15207333_

Round 1

Reviewer 1 Report

This article investigated the influence of milling time on the phase composition and structure of two kinds of Mg-Zn-Ca-Ag alloys only without involving the study on mechanical properties and strength of materials. Some comments are as follows:

1. Are the images of SEM, shown in Figs.3  and 4,  only the accumulation of particles of these four different elements just after milling or the one with grains and grain boundarys? How about the void ratio of these porous materials?

2. Why the amorphous process was observed only in the case of Mg65Zn30Ca4Ag1 alloy after milled 50 and 70 hrs? It is reqired to indicate this amorphous phase in the figure.

3. As shown in Fig.2, why the phenomenon of higher peaks occur with milling time 30 hrs?

4. As shown in Table 1, the element of alloy should be Ag instead of Au?

5. Does this study perform the process of Plasma Spark Sintering after milling of powders?

6. Several errors in wording should be carefully checked and revised.

Author Response

Dear Reviewer,

Thank you very much for your suggestions. All of them have been taken into consideration in order to improve this paper.

Please note that the referees’ comments are given in italic and my replies are given in bold. The changes in the text of the revised manuscript are highlighted in red.

All my responses to your comments are highlighted using the "Track Changes" function in Microsoft Word.

Q1: Are the images of SEM, shown in Figs.3  and 4,  only the accumulation of particles of these four different elements just after milling or the one with grains and grain boundarys? How about the void ratio of these porous materials?

Our reply: Thank you for this question.

Figures 3 and 4 show an example of the microstructure (SEM) and the results of the chemical analysis (EDS) of the Mg65Zn30Ca4Ag1 alloys after 70 h of milling time during the mechanical alloying. In this process can be formed new phase with grain boundaries (area 2 in Fig. 4). Chemical analysis shows that the homogeneous grains (1) visible in the images represent the alloy after mechanical synthesis with a chemical composition corresponding to the nominal one, and the grains of the heterogeneous microstructure (2) are richer in zinc with a simultaneous deficiency of Mg and Ca. This may indicate the precipitation of Mg-Zn phases, which is confirmed by X-ray examinations.

In materials produced by a mechanical alloying method, we do not determined porosity, which is used in sintered or compacted materials (in the pre-sintered stage).

Q2: Why the amorphous process was observed only in the case of Mg65Zn30Ca4Ag1 alloy after milled 50 and 70 hrs? It is required to indicate this amorphous phase in the figure.

Our reply: We thank the Reviewer for this comment. The amorphization is a progressing process. It is induced both thermally and mechanically. It is already observed in the initial and intermediate stages of grinding. However, after 50 and 70 hours, the amorphous phase is the dominant phase in the Ag1 alloy. It was found on the basis of the XRD results, where the disappearance of the reflections originating from the crystalline phases was observed (a drastic decrease in the intensity of the reflections). In the Ag 2 alloy is similar, but the disappearance of reflections takes place after 70 hours of grinding time. To confirm the presence of the amorphous phase in the Ag1 alloy, after 50 and 70 hours of grinding, an additional experiment was performed using a Transmission Electron Microscope (TEM). The results of the analysis are included in the text.

Q3: As shown in Fig.2, why the phenomenon of higher peaks occur with milling time 30 hrs?

Our reply: Thank you. In the case of the Ag1 alloy, a gradual process of amorphous material can be observed. The first step in the formation of an amorphous phase after 20 hours of grinding leads to intermediate crystalline phases after 30 hours to re-amorphization after 50 hours of grinding. Similar milling processes are observed in the literature. Exemplary detailed considerations on transitional amorphous forms were presented in the work on the mechanical synthesis of Ni-Ti alloys. Explanation is included in the text.

Q4: As shown in Table 1, the element of alloy should be Ag instead of Au?

Our reply: Yes, sorry. There was an editorial mistake. It has been corrected in the text.

Q5: Does this study perform the process of Plasma Spark Sintering after milling of powders?

Our reply: Thank you. This paper  is just part of the extensive research work done in the project number 2017/27/B/ST8/02927 financed by the National Science Center, number 2017/27/B/ST8/02927. The results of tests of similar alloys obtained as a result of MA and then sintering by the SPS method have been published in [33,40, Lesz S., Hrapkowicz B., GoÅ‚ombek K, Karolus M., Janiak P. Synthesis of Mg-based alloys with rare-earth element addition by means of mechanical alloying, Bull. Pol. Acad. Sci. Tech. Sci.  2021, 69, 5,  e137586, https://doi: 10.24425/bpasts.2021.137586 and Hrapkowicz B., Lesz S., Karolus M., Garbiec D., WiÅ›niewski J., Rubach R., GoÅ‚ombek K., Kremzer M., Popis J. Microstructure and Mechanical Properties of Spark Plasma Sintered Mg-Zn-Ca-Pr Alloy. Metals 2022, 12, 375. https://doi.org/10.3390/met12030375]. However, after an in-depth analysis of the magnesium alloy with Ag addition, the material obtained in the process of mechanical synthesis was selected.

Q6: Several errors in wording should be carefully checked and revised.

Our reply: Thank you, the article has been corrected.

Thank you again for all your comments. I hope you appreciate the specific changes. I have made in response to these comments and that overall you feel that the main arguments and contribution are now much stronger.

Sincerely yours,

Sabina Lesz and Małgorzata Karolus

Reviewer 2 Report

The article is devoted to the study of the properties of magnesium-based alloys, which are widely used in the automotive, aviation and medical industries. In general, the presented study is quite interesting and promising not only from a fundamental point of view, but also from a further practical application. The article corresponds to the declared journal and can be accepted for publication in the future after the authors answer a number of questions that have arisen during its analysis.

1. The abstract needs to be improved, the authors should reflect in more detail the novelty and practical significance of the work.

2. The processes of long-term grinding can be accompanied by the processes of chipping balls and the formation of impurity inclusions; the authors should clarify whether impurities were observed in their samples after mixing.

3. The authors should explain how exactly the crystal lattice parameters were calculated with such a high measurement accuracy and determination of the distortion value.

4. In the presented form, this material is more reminiscent of an analytical report, especially in the sections of determining the morphological features of the structures under study using SEM methods. The authors should pay more attention to the description of the observed effects and their interpretation.

5. The presented diffraction patterns are of very poor quality, the authors should significantly improve their quality and make them more visual.

Author Response

Dear Reviewer,

Thank you very much for your suggestions. All of them have been taken into consideration in order to improve this paper.

Please note that the referees’ comments are given in italic and my replies are given in bold. The changes in the text of the revised manuscript are highlighted in red.

All my responses to your comments are highlighted using the "Track Changes" function in Microsoft Word.

Q1: The abstract needs to be improved, the authors should reflect in more detail the novelty and practical significance of the work.

Our reply: Thank you, the information you requested has been added to the text. We modified the Abstract.

Q2: The processes of long-term grinding can be accompanied by the processes of chipping balls and the formation of impurity inclusions; the authors should clarify whether impurities were observed in their samples after mixing.

Our reply: I thank the Reviewer for this comment.  In fact, the problem of contamination during long grinding times is serious and frequently arises in this type of research. However, the authors, according to the X-ray analysis results presented in the article and in additionally conducted EDS analyzes using SEM and TEM measurements, did not observe the presence of impurities resulting from abrasion of the material of the balls and vessel walls. The information you requested has been added to the text.

Q3: The authors should explain how exactly the crystal lattice parameters were calculated with such a high measurement accuracy and determination of the distortion value.

Our reply: Thank you, the information you requested has been added to the text. Calculations of the parameters of unit cells of the identified phases and the determination of their crystallite size and lattice strains were performed using the Rietveld analysis, which is based on modifying the assumed crystallographic structure and analyzing the profile of the registered diffraction lines. The quality of the match is determined by the so-called R-parameters, which, in relation to the presented results, reached acceptable values below 10%. According to the literature, the generally accepted accuracy of such determined lattice parameters is at the level of 10-5 Å. The accuracy of crystallite size estimation, according to the Williamson-Hall theory, included in the Rietveld procedure, is 10-15%.

Q4: In the presented form, this material is more reminiscent of an analytical report, especially in the sections of determining the morphological features of the structures under study using SEM methods. The authors should pay more attention to the description of the observed effects and their interpretation.

Our reply: Your suggestions have been considered and included in the manuscript. According to the Reviewer’s comments, the discussion was changed.

For example:

“As compared to [48], the material presented in Fig. 3b resembles “plate” structure. In the referenced literature harder particles are being processed in the mechanical synthesis, while in case of this manuscript, silver is much more ductile in comparison. The outcome is different, clearer structure as shown in Figure 3.

The resulting structure appears due to the processes occurring due to the interaction between softer and harder particles during milling. The harder particles are breaking and decreasing in size, and simultaneously ductile, softer particles are strained and fractured into smaller bits. The structure occurs due to melting of the softer particles, trapping the harder ones inside and consolidating. The result is a homogeneous distribution of elements in the particle. Those claims are backed up by the results of the EDS analysis showcased in Figure 4. While section 1 features a uniform distribution of a smaller particulate after repeated processes, section 2 of the figure shows an intermediate process, where the components of the particle are still in a transient phase [35].”

Q5: The presented diffraction patterns are of very poor quality, the authors should significantly improve their quality and make them more visual.

Our reply: I thank the Reviewer#1 for this comment. The figures has been improved in text.

Thank you again for all your comments. I hope you appreciate the specific changes. I have made in response to these comments and that overall you feel that the main arguments and contribution are now much stronger.

Sincerely yours,

Sabina Lesz and Małgorzata Karolus

Round 2

Reviewer 1 Report

Nil.

Reviewer 2 Report

The authors answered all the questions, the article can be accepted for publication.